# Idiopathic Anaphylaxis? Analysis of Data from the Anaphylaxis Registry for West Pomerania Province, Poland

**DOI:** 10.3390/ijerph192416716

**Published:** 2022-12-13

**Authors:** Iwona Poziomkowska-Gęsicka

**Affiliations:** Clinical Allergology Department, Pomeranian Medical University (PMU) in Szczecin, 70-111 Szczecin, Poland; iwona.poziomkowska@op.pl or i.poziomkowska@spsk2-szczecin.pl

**Keywords:** idiopathic anaphylaxis, molecular diagnostics, causes of anaphylaxis

## Abstract

The most common causes of anaphylaxis, according to various authors and depending on the age of the studied groups, are: Hymenoptera venom, food, and medications. Unfortunately, we are not always able to indicate the cause of anaphylaxis. There are data in the literature where as many as 41% of all cases are idiopathic anaphylaxis. Since the introduction of new diagnostic methods such as molecular diagnostics (MD) in our centre, the percentage of idiopathic anaphylaxis in the Anaphylaxis Register has significantly decreased. The purpose of this study was to identify possible causes of idiopathic anaphylaxis in patients with a history of moderate to severe anaphylactic reactions. After using MD, the causative agent was found in another 29 people. The proportion of people with idiopathic anaphylaxis in the Registry decreased from 9.2% to 3.5%. There were no significant differences in the incidence, although men appear to be slightly more common in primary idiopathic anaphylaxis. The mean age of primary idiopathic anaphylaxis was 40 years, but this was as high as 51 for anaphylaxis with alpha-gal allergy. Exercise may or may not be present as a cofactor despite its established role, e.g., in wheat-dependent exercise-induced anaphylaxis (WDEIA). In most of the analyzed cases, i.e., 70%, the reaction took place within an hour. The longest time interval from exposure to the development of symptoms is in the case of alpha-gal allergy; in this analysis, it was at least 5 h after ingestion of the so-called “red meat”. Patients are not aware of the disease, or further attacks cannot be prevented. As many as 80% had idiopathic anaphylaxis prior to visiting the centre, and 80% developed anaphylaxis after visiting the centre, which emphasizes the need to not stop the medical team in their search for the causes. As many as 93% of cases required medical intervention, of which adrenaline was used only in 34.5%, antihistamines in 86%, systemic glucocorticosteroids (sCS) in 75%, and fluids in 62% of cases. A total of 83% of patients received an emergency kit for self-administration. Idiopathic anaphylaxis can be resolved as known-cause anaphylaxis after a thorough medical history and, if possible, without exposing the patient after using appropriate, modern in vitro diagnostic methods, including molecular diagnostics. The diagnosis of idiopathic anaphylaxis should extend the diagnosis to include alpha-gal syndrome, LTP syndrome and WDEIA.

## 1. Introduction

The term ‘anaphylaxis’ is over 120 years old, and it was used for the first time in 1902 by Portier and Richet to describe a reaction opposite to prophylaxis [1]. At present, numerous definitions of anaphylaxis are available, although some of them are used more often than others.

Different definitions of anaphylaxis have been proposed since the first use of this term, as can be seen in Table 1. Contemporary definitions are presented in a paper by Turner in 2019 [2].

Due to the advancement of knowledge and research, we know that anaphylactic reactions after exposure to a causative factor may be delayed. It does not have to be an allergic reaction, and it does not occur every time following exposure to a given substance. Generally, anaphylaxis is most commonly defined as an acute, severe, potentially life-threatening systemic hypersensitivity reaction [3] and remains a clinical diagnosis. Most healthcare professionals define anaphylaxis as a serious, generalized, allergic, or hypersensitivity reaction that can be life-threatening and even fatal [4,5,6,7,8]. A useful, helpful laboratory test to confirm anaphylaxis, whether idiopathic, drug-induced, or insect sting-related, is to detect elevated tryptase levels within 2–3 h of the reaction.

Based on various data, the most common causes of anaphylaxis include Hymenoptera venom, foods, and medications, depending on the age of the study groups [9]. Unfortunately, we cannot always find the cause of anaphylaxis, and according to some data, idiopathic anaphylaxis accounts for up to 41% of all cases [10]. Since new diagnostic methods were introduced in our centre in 2016, the idiopathic anaphylaxis rates in the Anaphylaxis Registry have decreased significantly. Molecular diagnostics (MD), the new method mentioned above, helps not only to determine what insect caused the anaphylactic reaction (a wasp or a bee, which is important for the planning of immunotherapy) but can also find the causes of anaphylaxis that previously were considered to be idiopathic. It usually applies to food allergies, as in this case, certain molecules important for identifying the causes of anaphylactic reaction are underrepresented with the use of standard methods: commercial skin prick tests (SPT) and blood tests for allergen-specific immunoglobulins E (sIgE). Many publications, apart from the analysis of causes of anaphylaxis, present information about patient-specific risk factors and cofactors amplifying anaphylactic reactions [11,12,13,14,15,16,17,18,19]. Multiple episodes of anaphylaxis following the consumption of unconnected foods should raise concerns about the possibility of a hidden allergen-induced or “summation anaphylaxis” due to cofactor influence [20]. Therefore, it is important to pay attention to the specific circumstances in which anaphylaxis occurred in order to find the causative factor, e.g., nocturnal anaphylaxis may be associated with an allergy to mammalian red meat.

Wheat (*Triticum aestivum*) is one of the most common foods. Its primary ingredient (gluten) contributes to the development of autoimmune diseases [21,22], and from the point of view of an allergist, wheat may trigger primary food allergies in early childhood [23,24,25]. Over the last decades, cases of wheat-dependent exercise-induced anaphylaxis (WDEIA) have been reported. It is noteworthy that in this type of anaphylaxis, ingestion of the allergen may precede the anaphylactic reaction by a few hours, and physical exercise is a cofactor [26]. Although it has been established that the main allergen responsible for this type of reaction is omega-5-gliadin (Tri a 19) [27], anaphylaxis may also be caused by other wheat allergens [28]. In order to identify an allergy to omega-5-gliadin and associate it with clinical symptoms, molecular diagnostics (MD) is required. Another interesting group of proteins, molecules undetectable with the use of routine tests used in allergology, includes lipid transfer proteins (LTPs). These are plant panallergens, clinically significant primarily as food allergens. The clinical manifestations of hypersensitivity to LTP are characterised by a varied course, from mild local symptoms, limited to the oral cavity, skin or gastrointestinal tract, to anaphylactic shock, even in the same patient [29]. LTP concentration depends on the maturity, storage and growing conditions of the fruit [30]. LTP has been identified in many fruit and vegetables, in particular those from the rose family—including peaches (Pru p 3) and apples (Mal d 3)—but also in hazelnuts (Cor a 8); in walnuts (Jug r 3); in the bean family (Lat. Fabaceae), e.g., in peanuts (Ara h 9); and many others [31]. It has also been explained that approximately 1/3 of patients with LTP allergy demonstrate good tolerance of peeled vegetables and fruit due to a higher allergen distribution in the peel. However, the risk of anaphylaxis still exists in this group of patients, especially in the presence of cofactors of allergic reactions, and consumption of peeled fruit and vegetables cannot be considered completely safe [29]. In 2007, Asero et al. were looking for fruit and vegetables safe for patients with LTP syndrome, and they determined that carrots, potatoes and melons have the lowest relative LTP concentrations and can be tolerated by the majority of patients [32,33]. In 2009, Commins and Platts-Mills presented a new disease syndrome in the form of delayed-onset anaphylaxis following the ingestion of red meat mediated by sIgE reaction to an oligosaccharide due to a tick bite [34]. The oligosaccharide is expressed in the cells of non-primate mammals, and it is present in the organs and meat of cows, pigs and lambs, as well as in feline epithelium. Direct contact with the epitope results in the production of antibodies against galactose-alpha-1,3-galactose (α-Gal). The identified antibody directed against the mammalian oligosaccharide epitope is associated with two distinct anaphylactic responses:An immediate onset reaction—anaphylactic shock during the first exposure to intravenous cetuximab;A delayed onset reaction—occurring 3–6 h after ingestion of mammalian meat, e.g., beef, pork or offal.

Symptoms occur primarily following the consumption of mammalian meat and less often after the ingestion of products containing gelatin [35]. Although pork kidney contains the most alpha-gal epitopes, any other product containing even small quantities of this allergen may cause a reaction [34,36] ranging from urticaria to angioedema or delayed anaphylaxis.

The usefulness of molecular diagnostics (MD) in identifying the causes of allergic anaphylaxis is evident, as this method helps to establish a diagnosis and a cause of anaphylaxis previously considered, e.g., to be idiopathic. This translates to specific clinical observations and provides information supporting the further management of patients [37]. Uncertain false sIgE results of the extract-based tests are due to, among other things, the presence of carbohydrate residues, so-called CCD (cross-reactive carbohydrate determinants), even in unrelated groups of products, although in most cases, sIgE results, both for extract-based tests and for commercial SPTs, are false-negative.

## 2. Objective

The objective of this study was to identify the possible causes of anaphylaxis primarily classified as idiopathic in patients with a history of moderate to severe anaphylactic reactions.

## 3. Materials and Methods

### 3.1. Study Design and Data Collection

The Anaphylaxis Registry of the Allergology Department PUM in Szczecin is a database collecting standardised empirical information about moderate and severe anaphylactic reactions (grades II–IV by Ring and Messmer classification [38]). The data of the patients reporting to the centre due to anaphylaxis experienced within the previous year are registered by physicians “after allergology”. The data were obtained from medical interviews and all the available medical records (including the records of emergency treatments, if available). Additional information about comorbidities and the repeatability of reactions was collected. Exercise as a cofactor of an anaphylactic reaction was analysed according to its intensity: mid intensity (e.g., walking), moderate-intensity (e.g., brisk walking or housework) or high-intensity (e.g., running, physical work).

The Ethical Committee of the Pomeranian Medical University in Szczecin was notified about the project, although its approval was not required, as the study is retrospective.

### 3.2. The Basic Questionnaire Was a Simplified Version of the Network for Online-Registration of Anaphylaxis Survey (NORA) from Berlin [9]; More Details in Other Publication [39] and Appendix B

The analysis included the cases submitted to the Anaphylaxis Registry from 2006 to 2021 that met the diagnostic criteria for moderate or severe anaphylaxis [39,40] (the scope of the registry).

### 3.3. Basal Measurement of Tryptase and sIgE

The levels of sIgE against rMald3 (LTP), rTri a 19 (omega-5-gliadin), and nGal-alpha-1,3 (thyroglobulin = α-Gal) were determined using the UniCap 100 Phadia/Sweden/ThermoFisher device.

### 3.4. Statistical Analysis

Obtained data were analyzed using the Statistica 12 software package (StatSoft, Inc., Cracow, Poland license, Tulsa, OK, USA). A basic statistics panel was used for data processing: descriptive statistics and, due to the qualitative features used in the analysis, nonparametric tests were used. For comparison between groups, the collected data was presented in the form of a multidivisional table; for qualitative variables, we used Pearson’s chi-squared test. Statistical significance was adopted at a *p*-value of *p* < 0.05 and *p* < 0.005.

## 4. Results

### 4.1. Descriptive Statistics

Of all 19,612 new patients examined at the Allergology Department in 2006–2021, there were 519 cases of moderate and severe anaphylaxis (grades II–IV by Ring and Messmer classification), which accounted for 2.64% of new patients.

The mean annual incidence of anaphylaxis in the population of Western Pomerania province in the analysed 16-year period was 0.0019% (0.002%).

#### 4.1.1. The Structure of the Research

Idiopathic anaphylaxis in the group of patients with confirmed anaphylaxis (N = 519) was observed in 47 cases, accounting for 9.2% of the anaphylactic reactions in the Registry.

In the group of patients (N_i_ = 47) with moderate and severe anaphylaxis (grades II–IV by Ring and Messmer classification) classified as idiopathic, in vitro studies using MD identified a causative factor in 29 patients—Appendix A. Thus, the percentage of patients with idiopathic anaphylaxis in the Registry decreased from 9.2% to 3.5%.

The following causes of anaphylaxis were identified:-In 15 cases, ingestion of wheat flour, and more specifically, of omega-5-gliadin;-In 8 cases, ingestion of fruit containing LTP;-In 6 cases, episodes of nocturnal anaphylaxis waking patients up were explained, as an α-Gal allergy to meat was found.

Further detailed analyses included these *n* = 29 individuals previously diagnosed with idiopathic anaphylaxis due to the imperfection of the diagnostic methods used or lack of patient consent to a challenge test. Following the diagnosis and identification of the underlying cause, the 29 patients did not experience any more episodes of anaphylaxis.

The allergist at the Allergology Department assessed the chance of developing mastocytosis based on REMA criteria. No diagnosis of mastocytosis was performed in the study group.

#### 4.1.2. The Gender in Analysed Group According to the Cause of Anaphylaxis (*n* = 29)—Shows Figure 1


Regarding gender distribution in the study group, anaphylaxis developed more often in males, but the difference was not statistically significant (*p* = 0.38). A noticeable difference in incidence can be observed for α-Gal (5 males vs. 1 woman), but statistical significance was not achieved. LTP-induced anaphylaxis was found with an equal frequency of 50% in men and women, with omega-5-gliadin observed in 8 males vs. 7 women. 

#### 4.1.3. Age at First Occurrence of Reaction in the Analysed Group—Can Be Seen in Figure 2

The mean age at the first occurrence of anaphylaxis in the study group was 40 years, ranging from 18 to 77 years, SD ± 15.5. Interestingly, in patients with anaphylaxis following the ingestion of red meat (alpha-gal allergy), the mean age was higher, 51 years SD ± 13.6, compared to allergy to LTP or omega-5-gliadin (35 and 38 years, respectively).

#### 4.1.4. The Place Where Reaction Occurs—Shows Figure 3

In the analysed group, only the alpha-gal allergy subgroup 100% (6 people) of anaphylactic reactions occurred at home. Anaphylaxis following the ingestion of wheat containing omega-5-gliadin, in most cases, occurred in places other than home (*p* < 0.05).

#### 4.1.5. Effort as a Cofactor, According to Its Intensity—Can Be Seen in Figure 4—Present or No and in Figure 5—Intensity of Effort 

In anaphylaxis classified as idiopathic, food is harmful to the patient, usually in the presence of a cofactor (an additional factor). The collected data indicate that for alpha-gal allergy, this is not necessary, as none of the patients in the study group required a cofactor was for the reaction to occur after the ingestion of red meat. Conversely, in the group of patients developing an anaphylactic reaction to wheat (omega-5-gliadin), in 87% of cases, a cofactor in the form of physical exercise was required (*p* < 0.05).

In the study group, the anaphylactic reaction was induced regardless of the type of exercise. Both mild and heavy exercise (24% for each type) were cofactors of anaphylaxis (*p* < 0.05), compared to 7% of cases in which moderate-intensity exercise was involved.

### 4.2. Comparison of the Cases of Anaphylaxis Previously Classified as Idiopathic, Considering Their Causes (after the Use of MD); Early and Late Reactions, Medical Interventions (n = 29)

#### 4.2.1. Time from Food Ingestion to Reaction Onset According to Triggering Factors—Shows Figure 6

The collected data indicate that the observation time of over 60 min from the ingestion of a given food to the symptom onset is not specific to alpha-gal allergy only but may also apply to reactions to LTP in up to 70% of cases. In the group of patients with an allergy to wheat (omega-5-gliadin), a statistically significant majority of anaphylactic reactions (*p* < 0.05) involves symptom onset within up to one hour.

#### 4.2.2. Patient Characteristics According to Previous Reactions. We Can Seen in Figure 7—Occurrence Both of Earlier and Later Episodes of Anaphylaxis

Surprisingly, the collected medical history revealed that the vast majority of patients, i.e., 80% of cases, experienced previous episodes of anaphylaxis, which indicates the need for educational activity to encourage patients to report to specialists.

#### 4.2.3. Patient Characteristics According to Later Reactions

Following an episode of idiopathic anaphylaxis affecting the patient’s awareness of the disease, further episodes were observed in approximately 80% of patients. This is probably due to the fact that the causes of anaphylaxis had not been yet identified or no preventive measures had been introduced.

#### 4.2.4. Treatment Applied during the Episodes in Individual Groups—Shows Figure 8

In the study group, approximately 45% of patients took medications on their own compared to 31% (*p* < 0.05) of patients who did not use medications on their own. In nearly a quarter of cases, patients were not sure what therapy was applied, did not remember if they self-medicated, or had no reference documentation available.

##### Was Intervention of Healthcare Professionals Necessary during the Episodes of Anaphylaxis in Individual Groups (*n* = 29)?

Over 90% of patients needed the assistance of healthcare professionals, i.a. of emergency medical services, hospital emergency departments or urgent primary healthcare centres, which demonstrates that the reactions were severe and required further observation and/or treatment.

##### In How Many Cases Did Emergency Medical Service Have to Be Called to the Site Where the Reaction Occurred?

According to the collected data, in over 75% of cases, emergency medical services had to be called, and medical personnel had to arrive at the place where the reaction occurred to provide the patient with professional assistance.

##### How Many Cases Required Hospital Treatment/Observation

A total of 59% of the analysed cases required hospital treatment/observation.

### 4.3. What Medications Were Administered during the Episode of Anaphylaxis? We Can Seen in Figure 9

The highest percentage of patients during anaphylaxis received antihistamine drugs, systemic CS, and fluid therapy. The lowest percentage, 34.5% (10 cases), received an injection of adrenaline.

### 4.4. Providing the Patient with Emergency Kit after an Episode of Anaphylaxis—Shows Figure 10

A total of 83% of patients in the analysed group received an emergency kit for self-administration that can be used by a primary healthcare provider or a physician at a hospital emergency department.

### 4.5. Results of Additional Tests for Basal Tryptase and sIgE—Can Be Seen in Table 2

When analysing individual cases, one notices a lack of correlation between high sIgE values and the occurrence of anaphylactic reactions. The sIgE values for the sIgE against the recombinant rTri a19 antigen and the recombinant rMal d3 antigen ranged from 0.4 IU/mL (class 1) to maximum values of over 100 IU/mL (class 6), as measured using the UNI CAP100 device. The basal tryptase values in the study group were normal.

### 4.6. Comorbidity in the Study Group—Shows Figure 11

Regarding the diseases concurrent with anaphylaxis in the analysed group, 86% of patients had urticaria. This suggests that episodes of urticaria diagnosed before the reported anaphylaxis could have been the first stage of anaphylaxis. It is typically diagnosed retrospectively during another episode. 

### 4.7. Time from Reaction Onset to Arrival at a Centre, Based on the Study Group

Diagnostics and patients’ visits at a medical centre. Based on the collected data, 72% of patients visited a specialist within a year, and after moderate or severe anaphylaxis, only 14% of the patients visited a specialist after 12 months. Only one patient came to a specialist centre after four years. In three cases (approximately 10%), a patient was under the supervision of a specialist centre for another condition, and anaphylaxis developed later.

## 5. Discussion

**Idiopathic anaphylaxis.** Idiopathic anaphylaxis is a challenge for specialists due to the unknown causes of the anaphylactic reaction. The question of whether another episode of a serious reaction could be prevented remains unanswered. New diagnostic tools, including MD, help to find the hidden cause, but it requires engagement and awareness from the medical personnel.

In this study, in the group of anaphylaxis cases recorded over a period of 16 years (*n* = 519), initially, 47 cases of idiopathic anaphylaxis were observed, accounting for 9.2% of the cases in the Registry. The literature data presents different rates of idiopathic anaphylaxis. It has been reported that idiopathic anaphylaxis accounts for 6.5% [9], 9.8% [41], 13.7% [42], 20% [8] and even 41% [10] of all anaphylaxis cases. After using MD, the percentage of patients with idiopathic anaphylaxis in the Registry decreased from 9.2% to 3.5%. The percentage of cases with identified causes (62% of the cases previously classified as idiopathic) in this analysis is higher compared to rates reported by other researchers [43,44]. When the syndrome of delayed anaphylaxis to red meat was described in 2009, the cases previously classified as “spontaneous” or “idiopathic” anaphylaxis were identified as anaphylaxis due to a known cause. In 2012, it became clear that in a large area of the Southern and Eastern USA, thousands of patients suffer from alpha-gal allergy [45]. Similarly, Pattanaik et al. [11] reported a significant decrease in the number of cases identified as idiopathic anaphylaxis (reduction from 59% to 35%), which can be explained by the number of diagnosed cases of alpha-gal allergy [46].

**Mean age.** The mean age in the study group was 40 years, which is similar to that reported in other sources: 34 years [47], 41.6 [48], and 44.2 [9] years. It is noteworthy that in this study, the mean age (51 years) for patients experiencing anaphylaxis after the ingestion of red meat (alpha-gal allergy) is considerably higher, possibly due to the fact that patients tolerate meat for years, and only a tick bite may induce an immune response and allergic anaphylaxis. This is the case for approximately 4% of sensitized individuals in the general population “control group” [49]. In a study by Tripathi et al., an anaphylactic response to alpha-gal was found in patients over 60 years of age [50]. It is equally important to note that this list also includes organs from mammals, such as the liver, intestine, heart and kidney. In addition to meat, other mammalian products include lard, suet, gelatin, pork rinds, and dairy products [50]. Most patients with IgE to alpha-gal will have positive IgE assays for milk but will nonetheless tolerate milk and dairy products. Gelatin has been reported to cause severe reactions either in the form of jelly candy and marshmallows or as intravenous preparations (Haemaccel and Gelofusine) [51].

**The place of reaction**. Regarding the place of reaction, from the patient’s point of view, it is irrelevant whether the reaction occurs at home or outside the house. However, if it takes place at night, in sleep or while resting, the literature data indicate that the symptoms typically occur between 23:00 and 2:00, and they should suggest an alpha-gal allergy [52]. In our register, 100% of reactions occurred at home only in this subgroup of patients with anaphylaxis initially classified as idiopathic. The question about the place where the reaction occurred may be of assistance, and thus, one should bear it in mind while collecting the medical history.

**The time interval between exposure and reaction.** The time interval between exposure and the onset of the reaction was also assessed. The analysed data indicate that the period of over 60 min between the ingestion of a given food and the symptom onset is specific not only for alpha-gal allergy but may also be observed in reactions to LTP proteins. The minimum time from the ingestion of red meat to the onset of reaction in the study group was 5 h, which demonstrates the variety of manifestations of alpha-gal allergy, as in the group of patients who developed a reaction after the consumption of pork kidney, the response typically occurred within 2 h following exposure [34,50]. The delayed onset of food-induced allergic reactions to alpha-gal may potentially be explained by the difference in time required for the epitope-rich molecules to enter the blood circulation. The digestion of glycoproteins and glycolipids rich in alpha-gal, followed by their resorption in chylomicrons and very low-density lipoproteins, may take a long time, even a few hours 3–7 [53].

**The presence of a cofactor**. In anaphylaxis initially classified as idiopathic, when we are looking for a causative factor, it appears that a given food is harmful only in specific circumstances, e.g., in the presence of a cofactor (an additional factor). In none of the patients in the study group was the presence of a cofactor necessary for a reaction to occur following the consumption of red meat, contrary to the study by Fischer et al. reporting the presence of cofactors, including alcohol and physical exercise [54]. In the group of patients who developed anaphylaxis after the consumption of wheat (omega-5-gliadin), in as many as 87% of cases, a cofactor in the form of physical exercise was required (resulting in WDEIA). The available literature data also demonstrate that reactions to the ingestion of omega-5-gliadin do not require physical exercise as a cofactor; it is not a necessary condition for the reaction to occur [55,56,57]. Exercise increases the intestinal absorption of allergens and hence the concentration of these substances in the blood [58,59,60]. Christensen et al. concluded that exercise lowers the threshold and increases the severity of the reaction to the food [55]. If exercise was identified as a cofactor, its type was of no consequence for the induction of the reaction. Both: mild and intensive exercise were cofactors of anaphylaxis in the same percentage of patients, compared to 7% of cases where moderate-intensity exercise was observed. The difference may result from problems with the definition of moderate-intensity exercise. Brockow et al. conducted challenge tests in 34 patients who had previously experienced anaphylaxis associated with exercise-induced allergy to omega-5-gliadin. In order to elicit objective symptoms, they performed challenge tests with the allergen alone or in the presence of cofactors: physical exercise, NSAID or ethanol. They demonstrated that physical exercise is not necessary to elicit a reaction and that NSAIDs or alcohol can be very effective cofactors of allergic reactions [57]. 

**The repeatability of the reaction**. Surprisingly, the medical history revealed that over 80% of patients had experienced similar episodes before. This demonstrates that educational activities are required for this group of patients. Healthcare professionals and patients should remember that each incident should be documented. This percentage is significantly higher than those reported in another publication assessing previous episodes of anaphylaxis in response to a drug or a Hymenoptera sting (34% and 37%, respectively) [61]. The data presented in this study indicate that the risk of another idiopathic anaphylactic reaction more than doubles when the trigger is unknown. On the other hand, after a confirmed idiopathic anaphylaxis that increased the awareness of the disease, another episode was also observed in approximately 80% of cases. This was probably due to the fact that the cause had not been identified, and preventive measures could not be undertaken.

**Medical help during anaphylaxis and anaphylaxis treatment**. Regarding the treatment of anaphylaxis initially classified as idiopathic, in the study group, approximately 45% of patients self-administered medications, compared to 31% of patients who did not use medications on their own. Over 90% of patients required interventions from healthcare professionals, for instance, emergency medical services, or had to stay in a hospital emergency department or urgent primary healthcare centre, which demonstrates that the reactions were severe and required further observation and/or treatment. According to the collected data, emergency medical services had to be called in over 75% of cases, and medical personnel had to arrive at the place where the reaction occurred to provide the patient with professional assistance. Unfortunately, the tendency to underuse adrenaline was observed in the study group. It was used in response to approximately 35% of anaphylactic reactions, similar to the Spanish data [62]—30.7% reported that they had used the auto-injector. The data from the NORA European Anaphylaxis Registry indicates that the percentage of adrenaline auto-injections or adrenaline use by emergency services at the place where the reaction occurred was only 14% for food-induced anaphylaxis, and it doubled to 28% following a Hymenoptera sting [9]. It appears that the main problem of many patients suffering from anaphylaxis is the fact that they have not been prescribed an auto-injector. Studies from Belgium demonstrated that despite adequate treatment of the acute condition, only 9% of patients with anaphylaxis were invited for allergological diagnostics or received an auto-injector with adrenaline/epinephrine [63]. Similar data were collected in 1995 in Munich; only 10% of cases received any information about further management or were given an emergency kit. In Germany, adrenaline is increasingly often used in severe reactions (grades III and IV), and adrenaline/epinephrine is given as the initial measure in 20% of cases [64]. Slightly different data were presented in a retrospective analysis from 2003, demonstrating that 29% of children with recurring anaphylaxis were treated with the use of an adrenaline auto-injector [65]. It is worth emphasising that the subsequent need for another adrenaline dose and for hospitalisation was reduced in the patients who had received an adequate adrenaline dose using an auto-injector during the anaphylactic episode [66]. The analysed data indicate that antihistamine medications (AH) were used in 86% of the analysed cases, which is a significantly higher rate than the one found in the European registry—53% [9], but similar to the data from Belgium, where 85% of cases received AH medications [63]. No high-quality evidence from randomized, controlled trials exists to support the use of H1-antihistamines and H2-antihistamines in the treatment of anaphylaxis [66,67]. Despite the lack of evidence, it is still recommended in guidelines for the management of anaphylaxis [68,69]. In the study group, systemic glucocorticoids were used for the treatment of anaphylaxis in 76% of cases, a percentage within the range determined by the data from Belgium, i.e., 89% [63], and from NORA, i.e., 60% [9]. There is no evidence from randomized, controlled trials to confirm the effectiveness of glucocorticoids in the treatment of anaphylaxis [70,71]. They potentially relieve protracted anaphylaxis symptoms and are thought to prevent biphasic anaphylaxis, although these effects have never been proven [68,72]. Volume substitution is a treatment of choice in anaphylaxis. In the study group, it was administered in only 62% of reactions. Supportive fluid therapy may also be used in severe anaphylaxis with respiratory symptoms if a second adrenaline dose by intramuscular injection is required [73].

**Providing the patient with a self-treatment kit**. As many as 83% of patients from the analysed group were provided with an emergency kit for self-administration, similar to the Belgian data, which reported that 87% of patients received an emergency kit [63]. This is a good indicator of the awareness of healthcare professionals. Most studies demonstrate that epinephrine auto-injectors are rarely prescribed in allergology clinics [8], although Australian studies [74,75] and an Italian study [76] revealed that auto-injectors were prescribed to 100% or nearly 100% of patients [77]. On the other hand, Sicherer et al. observed that epinephrine auto-injectors were prescribed only to 23% of adults and 46% of children following a general systemic reaction to nuts or peanuts [78].

**sIgE values.** The analysis of individual cases in this study demonstrates that high sIgE values are unrelated to the development of an anaphylactic reaction. Values of the sIgE against recombinant rTri a 19 antigen and recombinant rMal d3 antigen ranged from 0.4 IU/mL (class 1) to maximum values of over 100 IU/mL (class 6), as measured by the UNI CAP 100 device. They were exactly in the range of 0.5–38.8 IU/mL for omega-5-gliadin, 0.4—above 100 IU/mL for LTP, and 1.46—above 100 IU/mL for alpha-gal. According to the data available in the literature, for alpha-gal, a positive result is determined by an sIgE level of over 0.1 IU/mL [78], but often, the level is very high, and it exceeds 100 IU/mL [44,79,80].

In 2012, Pascal et al. [29] published an article concerning 45 cases of patients with LTP syndrome. They demonstrated that LTP-specific IgE concentrations do not correlate with the intensity of clinical symptoms.

**Comorbidity in the study group**. Regarding the comorbidities observed in the analysed group, 86% of patients had urticaria. It appears that urticaria diagnosed prior to the anaphylaxis reported in the register was actually the first stage of anaphylaxis. The pediatric population responds similarly, primarily with urticaria, to alpha-gal allergy [45]. The determination of alpha-gal sIgE should be considered both in children and adults with chronic urticaria, angioedema or idiopathic anaphylaxis, especially in regions where ticks are found [45]. Concluding the discussion of anaphylaxis in alpha-gal allergy, it is worth noting that anaphylactic reactions are not limited to late-onset episodes. Immediate reactions may develop in these patients following the administration of cetuximab, a vaccine against measles-mumps-rubella (MMR) [81], and a zoster vaccine, as they contain a lot of gelatin, which is abundant with alpha-gal epitopes [82].

**Diagnostics and patients’ visits to a medical centre.** Based on the collected data, 72% of patients visited a specialist within a year of the reaction, and 14% of patients visited a specialist after 12 months. In very rare cases, individual patients visited a specialist centre after four years. In three cases (approximately 10%), a patient was already under the supervision of a specialist centre due to other causes, and anaphylaxis occurred later, possibly due to the fact that the trigger factor had not been identified. The problem of anaphylaxis initially classified as idiopathic is important, as demonstrated by the facts. According to the literature data, the diagnosis of anaphylaxis in the course of alpha-gal allergy is delayed by 4 [83] to 10 years [44].

## 6. Limitations

Because this was a retrospective study, it may have been influenced by selection bias.

The small size of the analysed group demonstrating rare causes of anaphylaxis, including alpha-gal, LTP and omega-5-gliadin, is a significant limitation. As only individual cases involved cofactors other than physical exercise (menstruation, infection, alcohol, and NSAIDs), they were not analysed.

No paediatric cases were reported.

This is a small observational study and has many limitations. We did not perform a formal oral challenge test or skin prick tests with native meat, and not all the patients received commercial food tests, so they were not analysed in this study (moderate and severe anaphylaxis; therefore, the diagnostic programme started from in vitro tests).

## 7. Conclusions

Anaphylaxis is a severe and life-threatening reaction and is variable and unpredictable. The comprehensive management of patients who have had anaphylaxis should be complex, so a partnership between allergists, emergency medicine and primary care providers is necessary. The main reason for the investigation of a patient who has already experienced anaphylaxis is to uncover the causal antigen of an anaphylactic reaction. D\A diagnosis of idiopathic anaphylaxis should lead to the extension of diagnostics to alpha-gal syndrome, LTP syndrome or WDEIA. Idiopathic anaphylaxis can be solved as known-cause anaphylaxis after a careful medical history and, if possible, without exposing the patient after using appropriate, modern in vitro diagnostic methods, including molecular diagnostics (MD).

## Figures and Tables

**Figure 1 ijerph-19-16716-f001:**
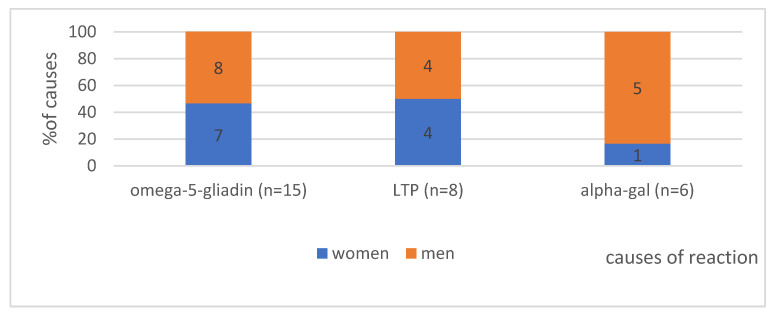
The gender distribution in analysed group with division into individual causes. Absolute values of numbers of patients are given in the diagram.

**Figure 2 ijerph-19-16716-f002:**
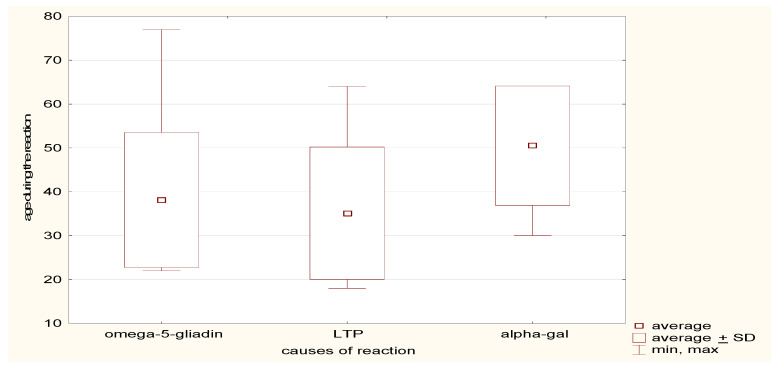
The age in analysed group—average, SD, minimum, and maximum, with division into individual causes.

**Figure 3 ijerph-19-16716-f003:**
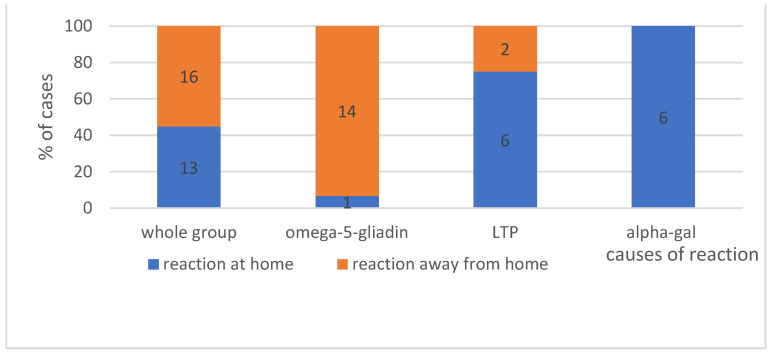
The place of reaction in analysed group, with division into individual causes. Absolute values of numbers of patients are given in the diagram.

**Figure 4 ijerph-19-16716-f004:**
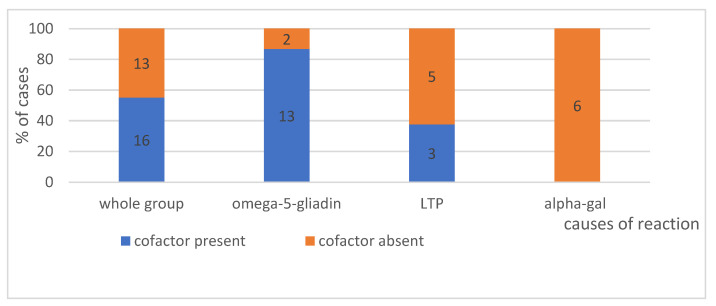
The presence of a cofactor in analysed group, with division into individual causes. Absolute values, no of patient-are given in the diagram.

**Figure 5 ijerph-19-16716-f005:**
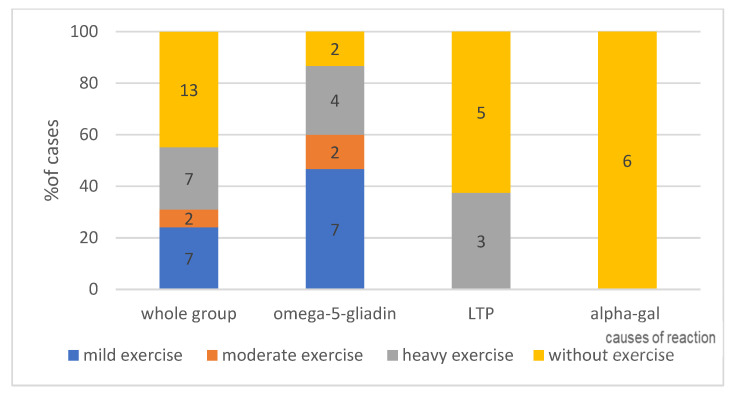
The exercise and intensity of exercise in analysed group, with division into individual causes. Absolute values of numbers of patients are given in the diagram.

**Figure 6 ijerph-19-16716-f006:**
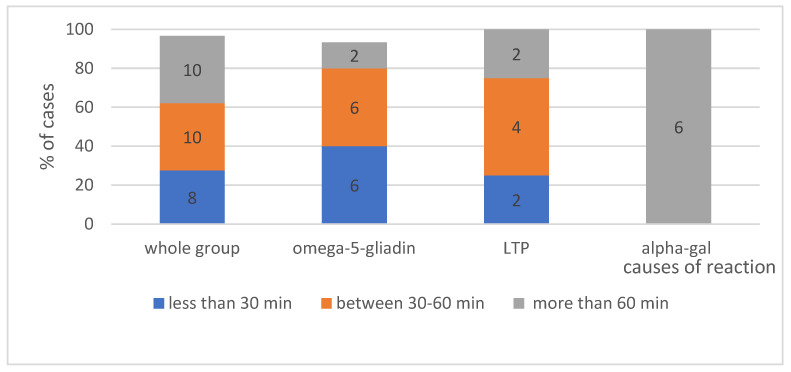
The time interval from exposure to the cause until the reaction occurs. Absolute values of numbers of patients are given in the diagram.

**Figure 7 ijerph-19-16716-f007:**
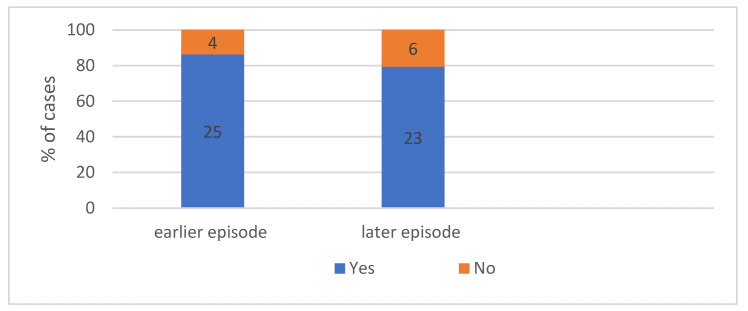
The percentage of earlier and later episodes in analysed group. Absolute values of numbers of patients are given in the diagram.

**Figure 8 ijerph-19-16716-f008:**
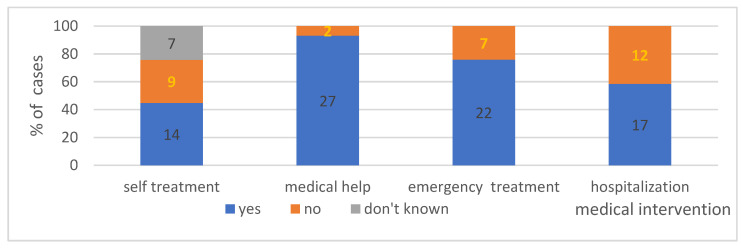
The medical intervention during the episode of anaphylaxis in analysed group. Absolute values of numbers of patients are given in the diagram.

**Figure 9 ijerph-19-16716-f009:**
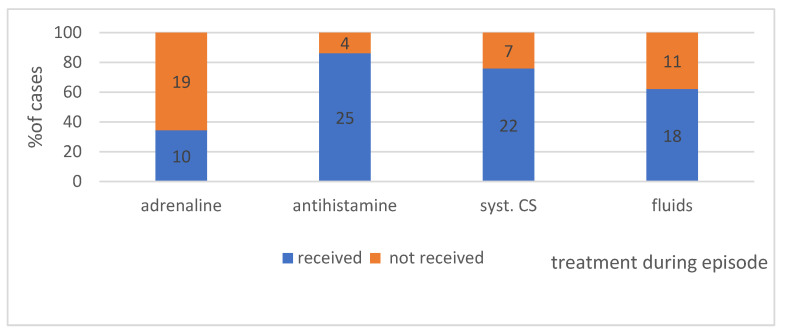
The treatment applied or lack thereof in the study group. Absolute values of numbers of patients are given in the diagram.

**Figure 10 ijerph-19-16716-f010:**
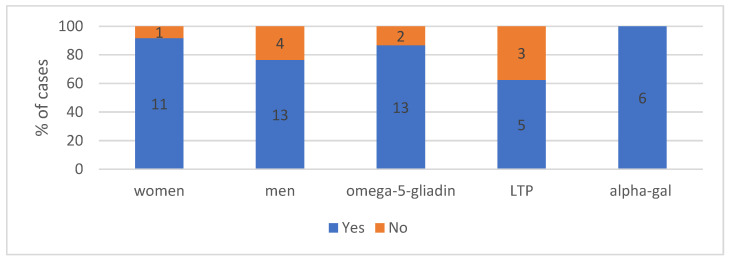
The provision of an emergency kit to patients for self-administration in the study group with division into individual causes and gender. Absolute values of numbers of patients are given in the diagram.

**Figure 11 ijerph-19-16716-f011:**
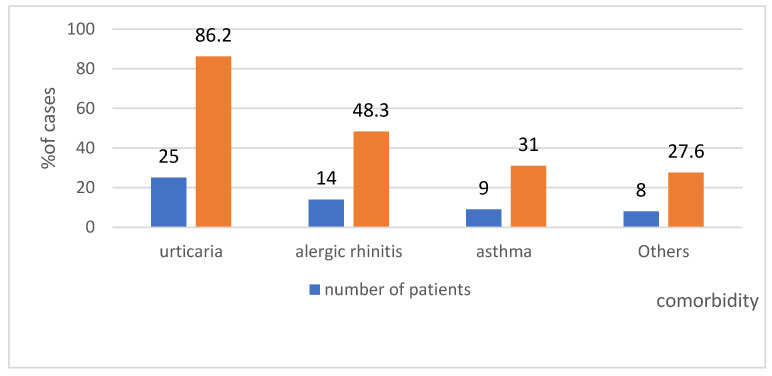
The comorbidity in the study group by % and numbers of patients.

**Table 1 ijerph-19-16716-t001:** Definitions of anaphylaxis.

WAO (Simons et al., 2011)	A serious life-threatening generalized or systemic hypersensitivity reaction. A serious allergic reaction that is rapid in onset and might cause death
EAACI (Panesar et al., 2013)	A severe life-threatening generalized or systemic hypersensitivity reaction. An acute, potentially fatal, multi-organ system allergic reaction.
AAAAI/ACAAI (Lieberman et al., 2010)	An acute life-threatening systemic reaction with varied mechanisms, clinical presentations, and severity that results from the sudden release of mediators from mast cells and basophils.
ASCIA (Brown et al., 2006)	Anaphylaxis is a serious, rapid-onset allergic reaction that may cause death. Severe anaphylaxis is characterized by life-threatening upper airway obstruction, bronchospasm and/or hypotension.

**Table 2 ijerph-19-16716-t002:** The characteristics of sIgE values and tryptase in the study group (*n* = 29) with division by gender.

	*n*	Average	SD	Min	Max
Tryptase	27	4.3	1.3	1.7	6.9
- Women	11	3.97	1.2	1.95	5.9
- Men	16	4.6	1.3	1.7	6.9
Omega-5-gliadin	15	9.5	10.7	0.5	38.8
- Women	7	9.7	9.1	0.5	26.9
- Men	8	9.3	12.6	0.5	38.8
LTP	8	19.8	33.8	0.4	100
- Women	4	25.4	49.7	0.4	100
- Men	4	14.3	10.6	3.93	23.9
Alpha-gal	6	54.9	43.4	1.46	100
- Women	1	1.46		1.46	1.46
- Men	5	65.6	38.8	8.1	100

## Data Availability

The data is archived on a server inaccessible from the outside, The research project is included and available for verification on the university website www.pum.edu.pl.

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
