# Peer review of "Idiopathic Anaphylaxis? Analysis of Data from the Anaphylaxis Registry for West Pomerania Province, Poland"

_ijerph, 2022, doi:10.3390/ijerph192416716_

Round 1

Reviewer 1 Report

Dear Author,

all in all - interesting manuscript. Anaphylaxis is always of interest. There is lack of novelity, it is just a retrospective report from a registry. Still it is current, it shows the spectrum of Polish patients and personally I find it inetresting.

There are some points that could be improved:

Figures -need editing, everything is merged together, poor quality of data presentation

The whole manuscript require extensive editing, inc, style and grammar - native speaker could help

The discussion is poorly written and require editing, to show the key points more clearly.

In summary - there is some merit in the work, but data presentation require improving.

Author Response

Thank you very much for Your time and honest review

Reviewer 2 Report

The manuscript is interesting since it provides information regarding the etiology of idiopathic anaphylaxis.

However I think more attention should be paid to mastocytosis (especially to indolent systemic mastocytosis) which may be responsible of idiopathic anaphylaxis even if basal tryptase is normal.

Did you investigate mastocytosis? If not, some information about this topic should be added in the manuscript.

Specific comments:

Line 19: change in "WDEIA (wheat-dependent exercise-induced anaphylaxis).

Line 24: .... the center, which emphasize

Page 2: detection of tryptase within 2-3 hours from the reactionis the only laboratory methos to confirm anaphylaxis. It should be clearly mentioned.

Line 132: add "." after allergology

Page 19, paragraph 4.6: I do not think chronic urticaria is a risk factor for any case of anaphylaxis. May be you mean acute urticaria attacks with no known causes?

Line 491: is that the title of the paragraph?

Author Response

(The authors gave the same response as above.)

Reviewer 3 Report

REVIEWER COMMENTS

COMMENTS FOR THE AUTHOR(S)

The manuscript titled “Idiopathic anaphylaxis? Analysis of data from the Anaphylaxis 2 Registry for West Pomerania province, Poland” is a paper focused on an important area of ​​contemporary allergology - anaphylaxis. The aim of the study was to identify the possible causes of anaphylaxis primarily classified as idiopathic in patients with a history of moderate to severe anaphylactic reactions.

Due to the use of molecular allergology, the number of patients with idiopathic anaphylaxis decreased from 9.2% to 3.5%. The author concluded that the diagnosis of idiopathic anaphylaxis should extend the diagnosis to include alfa-gal syndrome, LTP syndrome, or WDEIA.

The manuscript is quite interesting and takes into account the wide spectrum of studies results on idiopathic anaphylaxis in adults, but the way of presentation is not clear and in my opinion, it requires a few changes that could help deliver the ideas/views.

Major comments:

1.      That was a retrospective study based on The Anaphylaxis Registry of the Allergology Department PUM. The analysis included the cases submitted to the Anaphylaxis Registry from 2006 to 2021 and meeting the diagnostic criteria for moderate or severe anaphylaxis.

Has every patient been tested using molecular allergology (alfa-gal syndrome, LTP syndrome, or WDEIA) since 2006?   

2.      Do you mean wheat-dependent anaphylaxis or WDEIA (wheat-dependent exercise-induced anaphylaxis)?

3.      How was the diagnosis made?

The author rightly describes the lack of an oral food challenge test as a limitation of the study. It was a retrospective study, so how the significance of selected allergen components was determined?

4.      With information on pre- and post-anaphylaxis incidents (but at what point?) available, were the patients followed up prospectively? How many episodes of anaphylaxis have been observed in every patient?

5.      What symptoms were observed?

6.      There were differences between patients with idiopathic anaphylaxis and patients with known causes of anaphylaxis?

7.      Please add the results of the concentrations of sIgE to rTri a 19, rMal d 3, LTP and alpha-gal (not only classes - line 471).

8.      What was the number of patients with moderate and severe anaphylaxis?

9.      Were there any deaths in the analyzed group?

10.  Notes on statistics: add SD to mean, correct p-value (specific number instead of p<0.05)

11.  The analyzed group is the group of 47 patients or 29 patients? (for example line 223)
The study group…(line 249)

12.  Line 308-310: New diagnostic tools, including MD, help to find the hidden cause, but it requires engagement and awareness from the medical personnel. Please clarify this sentence.

13.  The discussion does not explain why, despite the fact that adrenaline was so rarely used, all patients survived…

14.  What about oxygen in the treatment?

Minor comments:

1.     Please add Figures to relevant places in the text.

2.     Please add numbers to percentages and vice versa.

3.     How was the contribution of cofactors studied?

4.     Please correct LPT into LTP.

5.     Please correct references (lines 50, 70, 72, 73, 76-80).

6.     What kind of LTP mattered in the studied patients?

7.     Please add a reference to line 130 - Ring and Messmer classification and line 137 – classification of exercise according to its intensity.

8.     This sentence needs correction: The basic questionnaire was a simplified version of the Network for Online-Registration of Anaphylaxis survey (NORA) from Berlin more information in other publication.

9.     There is no information on Appendix 1.

10. Line 161- what does it mean new patients?

11. Line 185 - due to the small size of this group – this is a part of the discussion, not the results.

12. Lack of p on Fig 2.

13. Please change Fig 2 into another graph - Chart of a frame-a moustache showing the average and median.

14. Figure 7: later or late?

15. Figure 8: Please add another graph.

16. Please standardize the charts.

17. Treatment of which episode of anaphylaxis was analyzed?

18. This is not the result - Line 265 - Anaphylaxis 265 usually requires a few medical interventions and/or medical follow-up.

19. Line 270 – the smallest? (add n and %).

20. What does mean volume? (Fluids?) - line 272.

21. Line 278 - It is a good indicator of the awareness of healthcare professionals. This is a part of the discussion.

22. Please correct the title of Table 1.

23. Line 294-296 (Healthcare professionals and patients should remember that a patient should receive a document recording each incident, and a referral to a specialist for further diagnostics.) This may be a part of the discussion.

24. Line 300-302: which episode of anaphylaxis was analyzed, if they had few?

25. Line 336 – add et al.

26. Line 491 - ? (subtitle?)

Author Response

Thank you very much for Your time and honest review.

Round 2

Reviewer 1 Report

Dear Author,

Still the figures are unacceptable for me. The subscriptions start sometimes with capital letter, sometimes not. The font is wrong. The titles do not go with Figure description. Tri a 19 is sometimes called "omega5gliadin" in different table "omega-5-gliadin". The quality of data presentation is poor. The disscussion is long and requires revision - it looses the merit of the work.

I find the information relevant, but impossible to print in this form. Please revise again.

Kind regards

Author Response

Thank you for your opinion, review.I standardized the charts, each time I corrected: omega-5-gliadin, capital letters in the charts removed, at the request of another reviewer, I added absolute numerical values.To make the results more readable, I shortened the discussion.I divided the discussions into subchapters, separate paragraphs, highlighted the analyzed issues in bold.

Reviewer 3 Report

Dear author,

This version is much better. Appendix 1 is still missing.

Author Response

Appendix 1

19 612

New patients of Allergology Department 

Thank You for Your comment, now its manuscript with appendix 1